# Demonstration-Guided Reinforcement Learning with Learned Skills

**Karl Pertsch,**[*] **Youngwoon Lee,**[†] **Yue Wu,** **Joseph J. Lim**[‡]

University of Southern California
https://clvrai.com/skild

**Abstract:** Demonstration-guided reinforcement learning (RL) is a promising approach for learning complex behaviors by leveraging both reward feedback and a set of target task demonstrations. Prior approaches for demonstration-guided RL treat every new task as an independent learning problem and attempt to follow the provided demonstrations step-by-step, akin to a human trying to imitate a completely unseen behavior by following the demonstrator's exact muscle movements. Naturally, such learning will be slow, but often new behaviors are not completely unseen: they share subtasks with behaviors we have previously learned. In this work, we aim to exploit this shared subtask structure to increase the efficiency of demonstration-guided RL. We first learn a set of reusable skills from large offline datasets of prior experience collected across many tasks. We then propose **Ski**ll-based **L**earning with **D**emonstrations (**SkiLD**), a demonstration-guided RL algorithm that leverages the provided demonstrations by following the demonstrated *skills* instead of the primitive actions, resulting in substantial performance improvements over prior demonstration-guided RL approaches. We validate its effectiveness on long-horizon maze navigation and robot manipulation tasks.

**Keywords:** Reinforcement Learning, Imitation Learning, Skill-Based Transfer

## 1 Introduction

Humans are remarkably efficient at acquiring new skills from demonstrations: often a single demonstration of the desired behavior and a few trials of the task are sufficient to master it [1, 2, 3]. To allow for such efficient learning, we can leverage a large number of previously learned behaviors [2, 3]. Instead of imitating precisely each of the demonstrated muscle movements, humans can extract the performed *skills* and use the rich repertoire of already acquired skills to efficiently reproduce the desired behavior.

Demonstrations are also commonly used in reinforcement learning (RL) to guide exploration and improve sample efficiency [4, 5, 6, 7, 8]. However, such demonstration-guided RL approaches attempt to learn tasks *from scratch*: analogous to a human trying to imitate a completely unseen behavior by following every demonstrated muscle movement, they try to imitate the *primitive actions* performed in the provided demonstrations. As with humans, such step-by-step imitation leads to brittle policies [9], and thus these approaches require many demonstrations and environment interactions to learn a new task.

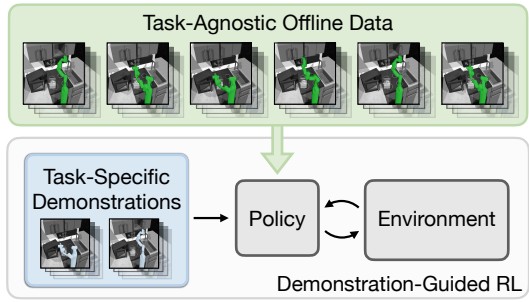

Figure 1: We leverage large, task-agnostic datasets collected across many different tasks for efficient demonstration-guided reinforcement learning by (1) acquiring a rich motor skill repertoire from such offline data and (2) understanding and imitating the demonstrations based on the skill repertoire.

[*]Correspondence to pertsch@usc.edu
[†]Work done during an internship at NAVER AI Lab
[‡]AI Advisor at NAVER AI Lab

5th Conference on Robot Learning (CoRL 2021), London, UK.

We propose to improve the efficiency of demonstration-guided RL by leveraging prior experience in the form of an offline "task-agnostic" experience dataset, collected not on one but across many tasks (see Figure 1). Given such a dataset, our approach extracts reusable skills: robust short-horizon behaviors that can be recombined to learn new tasks. Like a human imitating complex behaviors via the chaining of known skills, we can use this repertoire of skills for efficient demonstration-guided RL on a new task by guiding the policy using the demonstrated *skills* instead of the primitive actions.

Concretely, we propose **Ski**ll-based **L**earning with **D**emonstrations (**SkiLD**), a demonstration-guided RL algorithm that learns short-horizon skills from offline datasets and leverages them for following demonstrations of a new task. Across challenging navigation and robotic manipulation tasks this significantly improves the learning efficiency over prior demonstration-guided RL approaches.

In summary, the contributions of our work are threefold: (1) we introduce the problem of leveraging task-agnostic offline datasets for accelerating demonstration-guided RL on unseen tasks, (2) we propose SkiLD, a skill-based algorithm for efficient demonstration-guided RL and (3) we show the effectiveness of our approach on a maze navigation and two complex robotic manipulation tasks.

## 2   Related Work

**Imitation learning.**  Learning from Demonstration, also known as imitation learning [10], is a common approach for learning complex behaviors by leveraging a set of demonstrations. Most prior approaches for imitation learning are either based on behavioral cloning (BC, [11]), which uses supervised learning to mimic the demonstrated actions, or inverse reinforcement learning (IRL, [12, 13]), which infers a reward from the demonstrations and then trains a policy to optimize it. However, BC commonly suffers from distribution shift and struggles to learn robust policies [9], while IRL's joint optimization of reward and policy can result in unstable training.

**Demonstration-guided RL.** A number of prior works aim to mitigate these problems by combining reinforcement learning with imitation learning. They can be categorized into three groups: (1) approaches that use BC to initialize and regularize policies during RL training [6, 7], (2) approaches that place the demonstrations in the replay buffer of an off-policy RL algorithm [4, 5], and (3) approaches that augment the environment rewards with rewards extracted from the demonstrations [8, 14, 15]. While these approaches improve the efficiency of RL, they treat each task as an *independent* learning problem and thus require many demonstrations to learn effectively, which is especially expensive since a new set of demonstrations needs to be collected for every new task.

**Online RL with offline datasets.**  As an alternative to expensive task-specific demonstrations, multiple recent works have proposed to accelerate reinforcement learning by leveraging *task-agnostic* experience in the form of large datasets collected across many tasks [16, 17, 18, 19, 20, 21]. In contrast to demonstrations, such task-agnostic datasets can be collected cheaply from a variety of sources like autonomous exploration [22, 23] or human tele-operation [24, 25, 26], but will lead to slower learning than demonstrations since the data is not specific to the downstream task.

**Skill-based RL.** One class of approaches for leveraging such offline datasets that is particularly suited for learning long-horizon behaviors is skill-based RL [22, 27, 28, 29, 30, 31, 24, 32, 26, 33, 16]. These methods extract reusable skills from task-agnostic datasets and learn new tasks by recombining them. Yet, such approaches perform *reinforcement learning* over the set of extracted skills to learn the downstream task. Although being more efficient than RL over primitive actions, they still require many environment interactions to learn long-horizon tasks. In our work we combine the best of both worlds: by using large, task-agnostic datasets and a small number of task-specific demonstrations, we accelerate the learning of long-horizon tasks while reducing the number of required demonstrations.

## 3   Approach

Our goal is to use skills extracted from task-agnostic prior experience data to improve the efficiency of demonstration-guided RL on a new task. We aim to leverage a set of provided demonstrations by following the performed *skills* as opposed to the primitive actions. Therefore, we need a model that can (1) leverage prior data to learn a rich set of skills and (2) identify the skills performed in the demonstrations in order to follow them. Next, we formally define our problem, summarize relevant prior work on RL with learned skills and then describe our demonstration-guided RL approach.

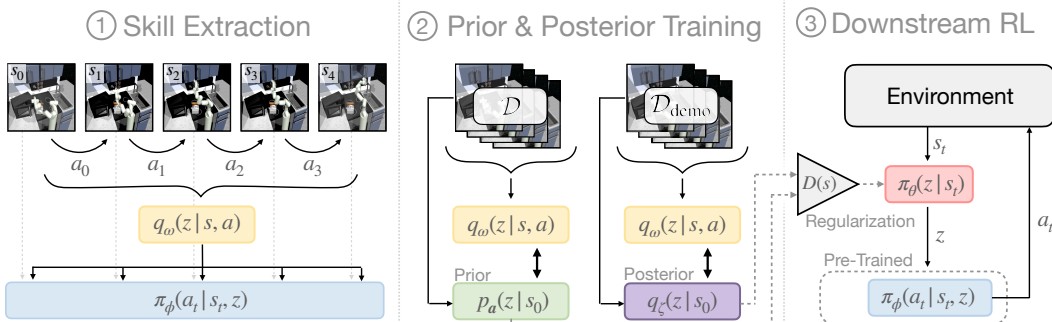

Figure 2: Our approach, SkiLD, combines task-agnostic experience and task-specific demonstrations to efficiently learn target tasks in three steps: (1) extract skill representation from task-agnostic offline data, (2) learn task-agnostic skill prior from task-agnostic data and task-specific skill posterior from demonstrations, and (3) learn a high-level skill policy for the target task using prior knowledge from both task-agnostic offline data and task-specific demonstrations. **Left**: Skill embedding model with skill extractor (**yellow**) and closed-loop skill policy (**blue**). **Middle**: Training of skill prior (**green**) from task-agnostic data and skill posterior (**purple**) from demonstrations. **Right**: Training of high-level skill policy (**red**) on a downstream task using the pre-trained skill representation and regularization via the skill prior and posterior, mediated by the demonstration discriminator $D(s)$.

## 3.1 Preliminaries

**Problem Formulation** We assume access to two types of datasets: a large task-agnostic offline dataset and a small task-specific demonstration dataset. The task-agnostic dataset $\mathcal{D} = \{s_t, a_t, ...\}$ consists of trajectories of meaningful agent behaviors, but includes no demonstrations of the target task. We only assume that its trajectories contain *short-horizon* behaviors that can be reused to solve the target task. Such data can be collected without a particular task in mind using a mix of sources, e.g., via human teleoperation, autonomous exploration, or through policies trained for other tasks. Since it can be used to accelerate *many* downstream task that utilize similar short-term behaviors we call it *task-agnostic*. In contrast, the task-specific data is a much smaller set of demonstration trajectories $\mathcal{D}_{\text{demo}} = \{s_t^d, a_t^d, ...\}$ that are specific to a single target task.

The downstream learning problem is formulated as a Markov decision process (MDP) defined by a tuple $(\mathcal{S}, \mathcal{A}, \mathcal{T}, R, \rho, \gamma)$ of states, actions, transition probabilities, rewards, initial state distribution, and discount factor. We aim to learn a policy $\pi_\theta(a|s)$ with parameters $\theta$ that maximizes the discounted sum of rewards $J(\theta) = \mathbb{E}_\pi \big[ \sum_{t=0}^{T-1} J_t \big] = \mathbb{E}_\pi \big[ \sum_{t=0}^{T-1} \gamma^t r_t \big]$, where $T$ is the episode horizon.

**Skill Prior RL** Our goal is to extract skills from task-agnostic experience data and reuse them for *demonstration-guided* RL. Prior work has investigated the reuse of learned skills for accelerating RL [16]. In this section, we will briefly summarize their proposed approach Skill Prior RL (SPiRL) and then describe how our approach improves upon it in the *demonstration-guided* RL setting.

SPiRL defines a skill as a sequence of $H$ consecutive actions $\boldsymbol{a} = \{a_t, \ldots, a_{t+H-1}\}$, where the skill horizon $H$ is a hyperparameter. It uses the task-agnostic data to jointly learn (1) a generative model of skills $p(\boldsymbol{a}|z)$, that decodes latent skill embeddings $z$ into executable action sequences $\boldsymbol{a}$, and (2) a state-conditioned prior distribution $p(z|s)$ over skill embeddings. For learning a new downstream task, SPiRL trains a high-level skill policy $\pi_\theta(z|s)$ whose outputs get decoded into executable actions using the pre-trained skill decoder. Crucially, the learned skill prior is used to guide the policy during downstream RL by maximizing the following divergence-regularized RL objective:

$$J(\theta) = \mathbb{E}_{\pi_\theta} \left[ \sum_{t=0}^{T-1} r(s_t, z_t) - \alpha D_{\text{KL}} \big( \pi_\theta(z_t|s_t), p(z_t|s_t) \big) \right]. \tag{1}$$

Here, the KL-divergence term ensures that the policy remains close to the learned skill prior, guiding exploration during RL. By combining this guided exploration with temporal abstraction via the learned skills, SPiRL substantially improves the efficiency of RL on long-horizon tasks.

## 3.2 Skill Representation Learning

We leverage SPiRL's skill embedding model for learning our skill representation. We follow prior work on skill-based RL [26, 19] and increase the expressiveness of the skill representation by replacing SPiRL's low-level skill decoder $p(a|z)$ with a closed-loop skill policy $\pi_\phi(a|s,z)$ with parameters $\phi$ that is conditioned on the current environment state. In our experiments we found this closed-loop decoder to improve performance (see Section C for an empirical comparison).

Figure 2 (left) summarizes our skill learning model. It consists of two parts: the skill inference network $q_\omega(z|s_{0:H-1}, a_{0:H-2})$ with parameters $\omega$ and the closed-loop skill policy $\pi_\phi(a_t|s_t, z_t)$. Note that in contrast to SPiRL the skill inference network is state-conditioned to account for the state-conditioned low-level policy. During training we randomly sample an $H$-step state-action trajectory from the task-agnostic dataset and pass it to the skill inference network, which predicts the low-dimensional skill embedding $z$. This skill embedding is then input into the low-level policy $\pi_\phi(a_t|s_t, z)$ for every input state. The policy is trained to imitate the given action sequence, thereby learning to reproduce the behaviors encoded by the skill embedding $z$.

The latent skill representation is optimized using variational inference, which leads to the full skill learning objective:

$$\max_{\phi,\omega} \mathbb{E}_q \left[ \underbrace{\prod_{t=0}^{H-2} \log \pi_\phi(a_t|s_t, z)}_{\text{behavioral cloning}} - \beta \big( \underbrace{\log q_\omega(z|s_{0:H-1}, a_{0:H-2}) - \log p(z)}_{\text{embedding regularization}} \big) \right]. \qquad (2)$$

We use a unit Gaussian prior $p(z)$ and weight the embedding regularization term with a factor $\beta$ [34].

## 3.3 Demonstration-Guided RL with Learned Skills

To leverage the learned skills for accelerating demonstration-guided RL on a new task, we use a hierarchical policy learning scheme (see Figure 2, right): a high-level policy $\pi_\theta(z|s)$ outputs latent skill embeddings $z$ that get decoded into actions using the pre-trained low-level skill policy. We freeze the weights of the skill policy during downstream training for simplicity.

Our goal is to leverage the task-specific demonstrations to guide learning of the high-level policy on the new task. In Section 3.1, we showed how SPiRL [16] leverages a learned *skill prior* $p(z|s)$ to guide exploration. However, this prior is task-agnostic, i.e., it encourages exploration of *all* skills that are meaningful to be explored, independent of which task the agent is trying to solve. Even though SPiRL's objective makes learning with a large number of skills more efficient, it encourages the policy to explore many skills that are not relevant to the downstream task.

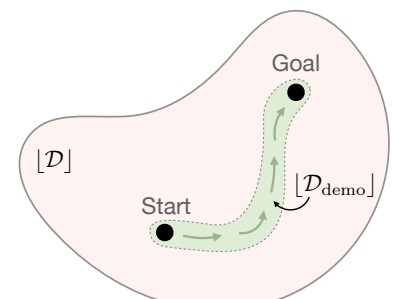

Figure 3: We leverage prior experience data $\mathcal{D}$ and demonstration data $D_{\text{demo}}$. Our policy is guided by the task-specific skill posterior $q_\zeta(z|s)$ within the support of the demonstrations (**green**) and by the task-agnostic skill prior $p_a(z|s)$ otherwise (**red**). The agent also receives a reward bonus for reaching states in the demonstration support.

In this work, we propose to extend the skill prior guided approach and leverage target task demonstrations to additionally learn a *task-specific* skill distribution, which we call *skill posterior* $q_\zeta(z|s)$ with parameters $\zeta$ (in contrast to the skill prior it is conditioned on the target task, hence "posterior"). We train this skill posterior by using the pre-trained skill inference model $q_\omega(z|s_{0:H-1}, a_{0:H-2})$ to extract the embeddings for the skills performed in the demonstration sequences (see Figure 2, middle):

$$\min_\zeta \mathbb{E}_{(s,a) \sim \mathcal{D}_{\text{demo}}} D_{\text{KL}} \big( q_\omega(z|s_{0:H-1}, a_{0:H-2}), q_\zeta(z|s_0) \big), \qquad (3)$$

A naive approach for leveraging the skill posterior is to simply use it to replace the skill prior in Equation 1, i.e., to regularize the policy to stay close to the skill posterior in every state. However, the trained skill posterior is only accurate within the demonstration support $\lfloor \mathcal{D}_{\text{demo}} \rfloor$, because by definition it was only trained on demonstration sequences. Since $|\mathcal{D}_{\text{demo}}| \ll |\mathcal{D}|$ (see Figure 3), the skill posterior will often provide incorrect guidance in states outside the demonstrations' support.

Instead, we propose to use a three-part objective that guides the policy to (1) follow the skill posterior *within* the support of the demonstrations, (2) follow the skill prior *outside* the demonstration support, and (3) reach states *within* the demonstration support. To determine whether a given state is within the support of the demonstration data we train a learned discriminator $D(s)$ as a binary classifier using samples from the demonstration and task-agnostic datasets, respectively.

In summary, our algorithm pre-trains the following components: (1) the low-level skill policy $\pi_\phi(a|s,z)$, (2) the task-agnostic skill prior $p(z|s)$, (3) the task-specific skill posterior $q_\zeta(z|s)$ and (4) the learned discriminator $D(s)$. Only the latter two need to be re-trained for a new target task.

Once all components are pre-trained, we use the discriminator's output to weight terms in our objective that regularize the high-level policy $\pi_\theta(z|s)$ towards the skill prior or posterior. Additionally, we provide a reward bonus for reaching states which the discriminator classifies as being within the demonstration support. This results in the following term $J_t$ for SkiLD's full RL objective:

$$J_t = \tilde{r}(s_t, z_t) \underbrace{-\alpha_q D_{\text{KL}}(\pi_\theta(z_t|s_t), q_\zeta(z_t|s_t)) \cdot D(s_t)}_{\text{posterior regularization}} \underbrace{-\alpha D_{\text{KL}}(\pi_\theta(z_t|s_t), p(z_t|s_t)) \cdot (1 - D(s_t))}_{\text{prior regularization}},$$

$$\text{with } \tilde{r}(s_t, z_t) = (1 - \kappa) \cdot r(s_t, z_t) + \underbrace{\kappa \cdot \left[ \log D(s_t) - \log \left( 1 - D(s_t) \right) \right]}_{\text{discriminator reward}}. \quad (4)$$

The weighting factor $\kappa$ is a hyperparameter; $\alpha$ and $\alpha_q$ are either constant or tuned automatically via dual gradient descent [35]. The discriminator reward follows common formulations used in adversarial imitation learning [36, 37, 8, 38].[4] Our formulation combines IRL-like and BC-like objectives by using learned rewards *and* trying to match the demonstration's skill distribution.

For policy optimization, we use a modified version of the SPiRL algorithm [16], which itself is based on Soft Actor-Critic [39]. Concretely, we replace the environment reward with the discriminator-augmented reward and all prior divergence terms with our new, weighted prior-posterior-divergence terms from equation 4 (for the full algorithm see appendix, Section A).

## 4 Experiments

In this paper, we propose to leverage a large offline experience dataset for efficient demonstration-guided RL. We aim to answer the following questions: (1) Can the use of task-agnostic prior experience improve the efficiency of *demonstration-guided* RL? (2) Does the reuse of pre-trained skills reduce the number of required target-specific demonstrations? (3) In what scenarios does the combination of prior experience and demonstrations lead to the largest efficiency gains?

### 4.1 Experimental Setup and Comparisons

To evaluate whether our method SkiLD can efficiently use task-agnostic data, we compare it to prior demonstration-guided RL approaches on three complex, long-horizon tasks: a 2D maze navigation task, a robotic kitchen manipulation task and a robotic office cleaning task (see Figure 4, left).

**Maze Navigation.** We adapt the maze navigation task from Pertsch et al. [16] and increase task complexity by adding randomness to the agent's initial position. The agent needs to navigate through a maze for hundreds of time steps using planar velocity commands to receive a sparse binary reward upon reaching a fixed goal position. We collect 3000 task-agnostic trajectories using a motion planner that finds paths between randomly sampled start-goal pairs. For the target task we collect 5 demonstrations for an unseen start-goal pair.

**Robot Kitchen Environment.** We use the environment of Gupta et al. [24] in which a 7DOF robot arm needs to perform a sequence of four subtasks, such as opening the microwave or switching on the light, in the correct order. The agent observes a low-dimensional state representation and receives a binary reward upon completion of each consecutive subtask. We use 603 teleoperated sequences performing various subtask combinations (from Gupta et al. [24]) as task-agnostic experience $\mathcal{D}$ and

---

[4]We found that using the pre-trained discriminator weights led to stable training, but it is possible to perform full adversarial training by finetuning $D(s)$ with rollouts from the downstream task training. We report results for initial experiments with discriminator finetuning in Section E and leave further investigation for future work.

separate a set of 20 demonstrations for one particular sequence of subtasks, which we define as our target task (see Figure 4, middle).

**Robot Office Environment.** A 5 DOF robot arm needs to clean an office environment by placing objects in their target bins or putting them in a drawer. It observes the poses of its end-effector and all objects in the scene and receives binary rewards for the completion of each subtask. We collect 2400 training trajectories by perturbing the objects initial positions and performing random subtasks using scripted policies. We also collect 50 demonstrations for the unseen target task with unseen object locations and subtask sequence.

We compare our approach to multiple prior demonstration-guided RL approaches that represent the different classes of existing algorithms introduced in Section 2. In contrast to SkiLD, these approaches are not designed to leverage task-agnostic prior experience: **BC + RL** initializes a policy with behavioral cloning of the demonstrations, then continues to apply BC loss while finetuning the policy with Soft Actor-Critic (SAC, [39]), representative of [6, 7]. **GAIL + RL** [8] combines rewards from the environment and adversarial imitation learning (GAIL, [13]), and optimizes the policy using PPO [40]. **Demo Replay** initializes the replay buffer of an SAC agent with the demonstrations and uses them with prioritized replay during updates, representative of [4]. We also compare our approach to RL-only methods to show the benefit of using demonstration data: **SAC** [39] is a state-of-the-art model-free RL algorithm, it neither uses offline experience nor demonstrations. **SPiRL** [16] extracts skills from task-agnostic experience and performs prior-guided

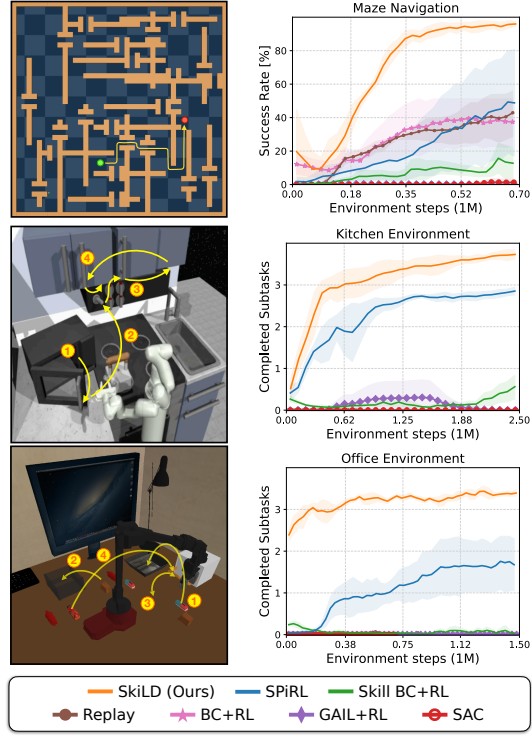

Figure 4: **Left**: Test environments, **top to bottom**: 2D maze navigation, robotic kitchen manipulation and robotic office cleaning. **Right**: Target task performance vs environment steps. By using task-agnostic experience, our approach more efficiently leverages the demonstrations than prior demonstration-guided RL approaches across all tasks. The comparison to SPiRL shows that demonstrations improve efficiency even if the agent has access to large amounts of prior experience.

RL on the target task (see Section 3.1)[5]. Finally, **Skill BC+RL** combines skills learned from task-agnostic data with target task demonstrations: it encodes the demonstrations with the pre-trained skill encoder and runs BC for the high-level skill policy, then finetunes on the target task using SAC. For further details on the environments, data collection, and implementation, see appendix Section B.

## 4.2 Demonstration-Guided RL with Learned Skills

**Maze Navigation.** Prior demonstration-guided RL approaches struggle on the task (see Figure 4, right) since rewards are sparse and only five demonstrations are provided. With such small coverage, behavioral cloning of the demonstrations' primitive actions leads to brittle policies which are hard to finetune. The Replay agent improves over SAC without demonstrations and partly succeeds at the task, but learning is slow. The GAIL+RL approach is able to follow part of the demonstrated behavior, but fails to reach the final goal (see Figure 8 for qualitative results). SPiRL and Skill BC+RL leverage task-agnostic data to learn to occasionally solve the task, but train slowly: SPiRL's learned, task-agnostic skill prior and Skill BC+RL's uniform skill prior during SAC finetuning encourage the exploration of many task-irrlevant skills[6]. In contrast, our approach SkiLD leverages the task-

---

[5]We train SPiRL with the closed-loop policy representation from Section 3.2 for fair comparison and better performance. For an empirical comparison of open and closed-loop skill representations in SPiRL, see Section C.

[6]Performance of SPiRL differs from Pertsch et al. [16] due to increased task complexity, see Section B.4.

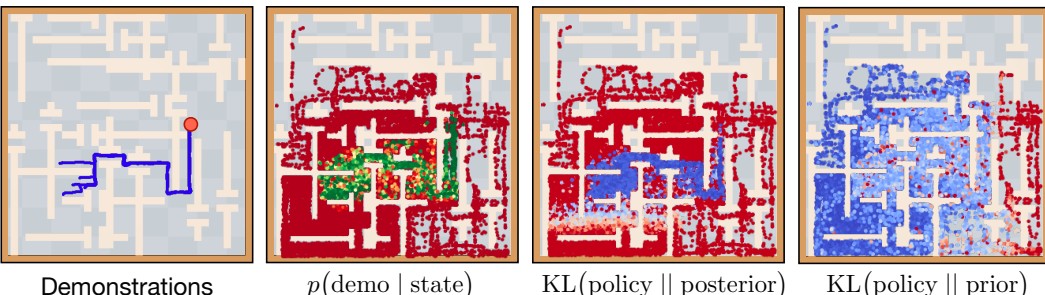

| Demonstrations | $p(\mathrm{demo} \mid \mathrm{state})$ | $\mathrm{KL}(\mathrm{policy} \parallel \mathrm{posterior})$ | $\mathrm{KL}(\mathrm{policy} \parallel \mathrm{prior})$ |

Figure 5: Visualization of our approach on the maze navigation task (visualization states collected by rolling out the skill prior). **Left**: the given demonstration trajectories; **Middle left**: output of the demonstration discriminator $D(s)$ (the **greener**, the higher the predicted probability of a state to be within demonstration support, **red** indicates low probability). **Middle right**: policy divergences to the skill posterior and **Right**: divergence to the skill prior (**blue** indicates small and **red** high divergence). The discriminator accurately infers the demonstration support, the policy successfully follows the skill posterior only within the demonstration support and the skill prior otherwise.

specific skill posterior to quickly explore the relevant skills, leading to significant efficiency gains (see Figure 5 for qualitative analysis and Figure 9 for a comparison of SkiLD vs. SPiRL exploration).

**Robotic Manipulation.** We show the performance comparison on the robotic manipulation tasks in Figure 4 (right)[7]. Both tasks are more challenging since they require precise control of a high-DOF manipulator. We find that approaches for demonstration-guided RL that do not leverage task-agnostic experience struggle to learn either of the tasks since following the demonstrations step-by-step is inefficient and prone to accumulating errors. SPiRL, in contrast, is able to learn meaningful skills from the offline datasets, but struggles to explore the task-relevant skills and therefore learns slowly. Worse yet, the uniform skill prior used in Skill BC+RL's SAC finetuning is even less suited for the target task and leads the policy to deviate from the BC initialization early on in training, preventing the agent from learning the task altogether (for pure BC performance, see appendix, Figure 13). Our approach, however, uses the learned skill posterior to guide the chaining of the extracted skills and thereby learns to solve the tasks efficiently, showing how SkiLD effectively combines task-agnostic and task-specific data for demonstration-guided RL.

## 4.3 Ablation Studies

In Figure 6 (left) we test the robustness of our approach to the **number of demonstrations** in the maze navigation task and compare to BC+RL, which we found to work best across different demonstration set sizes. Both approaches benefit from more demonstrations, but our approach is able to learn with much fewer demonstrations by using prior experience. While BC+RL learns each low-level action from the demonstrations, SkiLD merely learns to recombine skills it has already mastered using the offline data, thus requiring less dense supervision and fewer demonstrations. We also ablate the **components of**

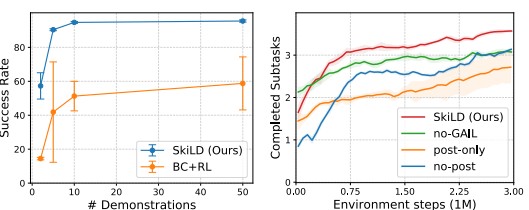

Figure 6: Ablation studies. We test the performance of SkiLD for different sizes of the demonstration dataset $|\mathcal{D}_{\mathrm{demo}}|$ on the maze navigation task (**left**) and ablate the components of our objective on the kitchen manipulation task (**right**).

**our RL objective** on the kitchen task in Figure 6 (right). Removing the discriminator reward bonus ("*no-GAIL*") slows convergence since the agent lacks a dense reward signal. Naively replacing the skill prior in the SPiRL objective of Equation 1 with the learned skill *posterior* ("*post-only*") fails since the agent follows the skill posterior outside its support. Removing the skill posterior and optimizing a discriminator bonus augmented reward using SPiRL ("*no-post*") fails because the agent cannot efficiently explore the rich skill space. Finally, we show the efficacy of our approach in the pure imitation setting, without environment rewards, in appendix, Section E.

---

[7]For qualitative robot manipulation videos, see `https://sites.google.com/view/skill-demo-rl`.

## 4.4 Robustness to Partial Demonstrations

Most prior approaches that aim to follow demonstrations of a target task, assume that these demonstrations show the *complete* execution of the task. However, we can often encounter situations in which the demonstrations only show incomplete solutions, e.g. because the agent's and demonstration's initial states do not align or because we only have access to demonstrations for a subtask within a long-horizon task. Thus, SkiLD is designed to handle such partial demonstrations: through the discriminator weighting it relies on demonstrations only within their support and falls back to following the task-agnostic skill prior otherwise. Thus it provides a robust framework that seamlessly integrates task-specific and task-agnostic data sources. We test

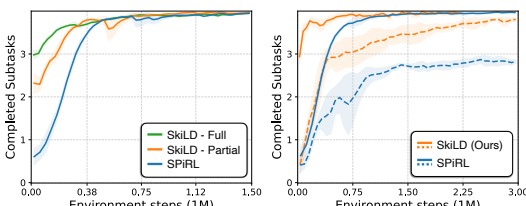

Figure 7: **Left**: Robustness to partial demonstrations. SkiLD can leverage partial demonstrations by seamlessly integrating task-agnostic and task-specific datasets (see Section 4.4). **Right**: Analysis of data vs. task alignment. The benefit of using demonstrations *in addition* to prior experience diminishes if the prior experience is closely aligned with the target task (**solid**), but gains are high when data and task are not well-aligned (**dashed**).

this experimentally in the kitchen environment: we train SkiLD with partial demonstrations in which we remove one of the subskills. The results in Figure 7 (left) show that "SkiLD-Partial" is able to leverage the partial demonstrations to improve efficiency over SPiRL, which does not leverage demonstrations. Expectedly, using the full demonstrations in the SkiLD framework ("SkiLD-Full") leads to even higher learning efficiency.

## 4.5 Data Alignment Analysis

We aim to analyze in what scenarios the use of demonstrations *in addition* to task-agnostic experience is most beneficial. In particular, we evaluate how the alignment between the distribution of observed behaviors in the task-agnostic dataset and the target behaviors influences learning efficiency. We choose two different target tasks in the kitchen environment, one with good and one with bad alignment between the behavior distributions (see Section F), and compare our method, which uses demonstrations, to SPiRL, which only relies on the task-agnostic data.

In the **well-aligned** case (Figure 7, right, solid lines), we find that both approaches learn the task efficiently. Since the skill prior encourages effective exploration on the downstream task, the benefit of the additional demonstrations leveraged in our method is marginal. In contrast, if task-agnostic data and downstream task are **not well-aligned** (Figure 7, right, dashed), SPiRL struggles to learn the task since it cannot maximize task reward and minimize divergence from the mis-aligned skill prior at the same time. Our approach learns more reliably by encouraging the policy to reach demonstration-like states and then follow the skill posterior, which by design is well-aligned with the target task.

In summary, our analysis finds that approaches which leverage both task-agnostic data *and* demonstrations, improve over methods that use either of the data sources alone across all tested tasks. We find that combining the data sources is particularly beneficial in two cases:

- **Diverse Task-Agnostic Data.** Demonstrations can focus exploration on task-relevant skills if the task-agnostic skill prior explores a too large set of skills (see Section 4.2).
- **Mis-Aligned Task-Agnostic Data.** Demonstrations can compensate mis-alignment between task-agnostic data and target task by guiding exploration with the skill posterior instead of the mis-aligned prior.

## 5 Conclusion

We proposed SkiLD, an approach for demonstration-guided RL that is able to leverage task-agnostic experience datasets *and* task-specific demonstrations for accelerated learning of unseen tasks. In three challenging environments SkiLD learns new tasks more efficiently than both, prior demonstration-guided RL approaches that are unable to leverage task-agnostic data, as well as skill-based RL methods that cannot effectively incorporate demonstrations. Future work should combine task-agnostic data and demonstrations for efficient learning in the real world and investigate domain-agnostic measures for data-task alignment to quantify the usefulness of prior experience for target tasks.

**Acknowledgments**

This research is supported by the USC Annenberg Fellowship, NAVER AI Lab, and NSF NRI-2024768. We would like to thank the lab members of the USC CLVR lab for valuable discussions throughout this project.

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
