# OpenReview forum: "Demonstration-Guided Reinforcement Learning with Learned Skills"
_robot-learning.org/CoRL/2021/Conference — CoRL2021 Poster_

### Official Review · Reviewer_fuNi · 2021-07-18

**Originality:** Fair
**Technical Quality:** Fair
**Clarity Of Presentation:** Good
**Impact:** 3

**Recommendation:**

Weak Reject: I recommend rejecting the paper, but will not argue for my recommendation if the majority of other reviewers have a different opinion.

**Summary:**

This work builds on top of SPiRL (Pertsch et al. 2020), with the insight that SPiRL encourages policies to explore many behaviors not relevant for the downstream task we actually want to solve. To address this, the authors propose a reasonable suggestion: fine-tune RL on the target task, but use the high-level skill spaces learned by SPiRL to benefit from the data efficiency and restricted search space.

The recipe is as follows: just as in SPiRL, first learn a VAE for short-horizon actions. This yields a skill posterior q(z|s,a), and a skill-conditioned action decoder pi(a|s,z) (which is later re-used as an “action decoder” for the policy). There is a small modification where the state is also passed to the “low-level policy”. One can also train the task-agnostic prior p(z|s) in this step, by predicting the inferred posteriors directly from state.

Next, *repeat* this for the target demo data, learning a task-specific skill posterior q(z|s). The task-agnostic prior p(z|s) and posterior q(z|s) are used in regularizing the high-level policy pi(z|s) being trained with RL. The full RL objective adds 3 distribution matching losses on top of the existing RL reward optimization: the raw reward is using a GAIL-style discriminator reward against target task demonstrations, encouraging the high level policy to match the posterior, and encouraging the policy to also lie within the prior (because the “skill posterior will often provide incorrect guidance outside of the demonstrations’ support”). Some of the loss coefficients are tuned automatically via dual gradient descent. Learned discriminator used to distinguish whether state is within support of the target task demonstration, and is thus used to tune which loss (posterior or prior regularization) matters more.

The authors evaluate this method on 3 simulated tasks, showing that it outperforms SPiRL and some other baseline algorithms (e.g. BC+RL) by a large margin.


**Issues:**

The fact that SPiRL and the proposed method are the only methods that do okay on the Kitchen and Office environments suggest to me that the baselines have not been tuned very well. For instance, it’s surprising that SAC and GAIL + RL do so poorly on the Maze Navigation environments.. Relay Policy Learning (Gupta ‘19 CORL) achieved high success rates on the kitchen environment using what essentially amounts to behavior cloning, which suggests that there is something sub-optimal in the baseline implementations of this method. I would like to see some convincing analysis as to why the baselines do so abysmally, given that other papers have reported decent goal-conditioned behavior cloning baselines.

I would like the authors to address the problem setting and data requirements in which this method is an ideal choice over other learning approaches, in particular ones that are simpler like GCBC or PPO brute-forced at a sim environment. For instance, is this done in the context of solving in sim and then transferring to real? How much target task data is available? What is a passable amount of overlap between the task-agnostic dataset distribution and the target task distribution?


**Reviewer Expertise:**

Good: General knowledge of the area

**Strengths And Weaknesses:**


Strengths:
1. Each of the pieces of the method (the use of high-level skills, the use of discriminator to detect OoD, the use of a task-agnostic skill prior to “catch” states outside of demonstration support, the use of a skill posterior to keep it close to the task-specific skill distribution) are reasonably motivated.
2. The re-use of various skill encoder components to form reward constraints and “loss switching” mechanisms is interesting.

Weaknesses:
1. Although the extensions to SPiRL presented here are interesting, there are too many moving parts to this method to make it practical. Depending on how one counts the pieces of a VAE encoder/decoder and GAN as one or several models, there are at least 4 models that need to be trained for this to work, each with their own set of hyperparameters. Furthermore, certain pieces are re-used in different places (e.g. the state discriminator is used for both reward shaping and switching between the penalizing the skill prior or skill posterior), which sounds like it makes it hard to debug if something is not working.
2. No real robot results. Tuning on-policy RL algorithms on real robots is challenging, and often SAC alone has enough hyperparameters to make things quite difficult and collect sufficient data. A demonstration that this method can be plausibly scaled to real robots is what I’d like to see.
3. A discriminator is used to switch between penalizing posterior / prior, but a discriminator in and of itself does not guarantee out-of-distribution errors. In fact, early on in training the state discriminator model may be misspecified.
4. It is not easy to assess the effectiveness or completeness of the latent skills until one trains the policy itself.
5. The paper’s admission that the task-specific skill posterior is not always accurate outside of the support of the task-specific demos seems like a major flaw in the method, requiring the user to bolt on several additional losses like the skill prior and discriminator loss. Going through this trouble of learning yet another couple models doesn’t imbue the robot with new capabilities beyond being able to solve the target task better, so it begs the question of whether one model can simply be tuned to solve the task well (for example, simply using PPO in simulation).



**Summary Of Recommendation:**

I do not spot major theoretical flaws with each of the components being proposed, but the method feels like 3 bolted-on learned components needed to get a specific idea ("fine-tune SPiRL with RL") to work well. It feels like to me that there can be a simplification of the method that makes it more practical for real robots.

---

> ### Author Response · Authors · 2021-08-23
> **Author Response (1/2)**
>
> Thank you for your detailed comments, we address your concerns below! We updated the manuscript and marked all changes made to address the reviewers’ comments in red.
>
> As a quick clarification, our paper goes beyond “finetuning SPiRL with RL” (SPiRL [1] already uses RL-based finetuning on the target task). Instead our method is designed to fuse skill-based learning and demonstration-guided RL (L41, alternative formulation: we aim to “augment SPiRL with demonstrations”).
>
> &nbsp;
>
> **When should I use this method over other learning approaches?**
>
> Our algorithm is designed to solve **complex, long-horizon tasks with sparse rewards** by jointly leveraging large task-agnostic experience and a small number of demonstrations. Sparse reward tasks are a common scenario in real-world robotic problems. In such tasks simpler approaches like vanilla RL (e.g. PPO) or simple imitation (e.g. BC) fail due to challenging exploration and accumulating errors over long horizons (Ross et al. 2011 [2]). More discussions on the shortcomings of such simpler approaches can be found in the original submission, L62-66.
>
> &nbsp;
>
> **What are the data requirements?**
>
> Our approach requires both task-agnostic data as well as task-specific demonstrations. Both datasets are collectable at scale in the real world (e.g. with human teleoperation -- Mandlekar et al. 2018 [3]). We assume that the task-agnostic data covers a wide variety of skills while the demonstrations show which skills are needed to solve the target task (see L98-105). We analyze the influence of overlap (aka alignment) between task-agnostic and demonstration data in detail in section 4.4 and, following your remarks, added a paragraph that summarizes the problem settings in which our algorithm is most beneficial in the updated manuscript (L351-364).
>
> &nbsp;
>
> **Is this done in the context of solving in sim and then transferring to real?**
>
> No, our algorithm aims to directly apply to real-world applications rather than sim-to-real approaches. The required data can be efficiently collected in the real world (e.g. through human teleoperation [3]), and is used to reduce the number of costly environment interactions while learning the target task, which addresses a major bottleneck for real-world applicability in prior works. See further discussion on real robot applicability below and in appendix, section G.
>
> &nbsp;
>
> **Baselines have low performance on test environments**
>
> We evaluate all algorithms on challenging, long-horizon problems with sparse rewards:
> - maze: 1k+ steps for successful task completion, only binary reward upon success
> - kitchen / office: requires precise control of high-DoF manipulator for 300-500 steps, sparse subtask completion reward
>
> These tasks are of substantially longer horizon than the typical evaluation tasks for LfD / demo-guided RL (e.g. single-object pick/place, door opening etc). We find that prior works that leverage demonstrations by imitating their primitive actions struggle on these long-horizon tasks.
>
> To concretely address the reviewer’s concerns: **in the maze environment** “vanilla” SAC has no hope of solving the exploration problem since it only receives reward upon successful completion of the 1k-step-horizon task. Our original submission explicitly analyzes the behavior of the GAIL+RL baseline in appendix, Fig. 8: we find that it successfully leverages the demonstrations to make progress towards the goal, but never reaches the final goal and thus never obtains the environment reward. We have verified our GAIL implementation on the common Mujoco benchmark tasks and extensively tuned the hyperparameters (update frequency and learning rate of policy and discriminator among others), but were not able to successfully imitate the long-horizon task from the small set of only 5 provided demonstrations.
>
> **In the kitchen environment** we are able to reproduce the imitation results of Gupta et al. [4] _before finetuning_: our BC baseline in Fig. 13 solves ~1.5 subtasks on average which is in line with the BC baselines of Gupta et al. w/o relabeling before finetuning (we do not need to perform goal-conditioned BC since the target task goal is fixed). We find that finetuning from this BC baseline (“BC+RL”) quickly deteriorates performance particularly in the more challenging robotics tasks that require precise control, since the uninitialized q-value function and SAC’s uniform prior over actions let the policy deviate from the BC initialization early on (in response to your remark we expanded our discussion on this in section 4.2 in the updated manuscript). Note that Gupta et al. additionally assume access to a dense low-level finetuning reward instead of just a sparse task-level reward, thus the results with finetuning are not comparable.

---

> > ### Author Response · Authors · 2021-08-23
> > **Author Response (2/2)**
> >
> > **Too many moving pieces, hard to tune?**
> >
> > All components of our approach except for the target task policy are trained offline using simple supervised learning, without the need for e.g. adversarial training or training in the RL loop. Thus, we find that the offline training of these components is robust to architecture and hyperparameter choices and their training results can be separately visualized and debugged -- without the need for costly RL experiments. Similar to prior work on skill-based learning (SPiRL, Pertsch et al. 2020 [1]) and maximum entropy RL mentioned by the reviewer (SAC), the main tuning for our method needs to be done on the weighting terms between reward and policy regularization during RL -- thus we do not believe that our method is significantly harder to tune than these prior works. Following the reviewers remarks, we have added a summary of the pre-trained components and a note about offline pre-training to section 3.3.
> >
> > &nbsp;
> >
> > **Not applicable to real robots**
> >
> > We discuss the applicability of our approach to real robotic systems in detail in appendix, section G. Our approach has multiple benefits over alternative RL approaches like “vanilla” SAC in terms of applicability to real robot systems. Most importantly, it performs most of the training **offline** from data that can be collected at scale in the real world (e.g. with human teleoperation -- Mandlekar et al. 2018 [3]). As a result it requires a lot fewer costly real-world environment interactions than alternative RL approaches. Its ability to leverage demonstrations can further improve sample efficiency. We evaluate our system on complex, long-horizon manipulation tasks with realistic high-DoF robotic agents across multiple state-of-the-art physics simulators (PyBullet & Mujoco). Thus we believe that the proposed algorithm has promise for real-world application, although we were not able to directly test on real robot systems due to the circumstances of the pandemic.
> >
> > &nbsp;
> >
> > **Skill-posterior doesn’t work outside demonstration support**
> >
> > The skill posterior was only trained on the demonstration data. Thus it is expected that it only provides reliable outputs within the support of its training data, i.e. within the demonstration support, analogous to how a BC policy only works within the demonstration support (Ross et al. 2011 [2]). We clarified this point in the updated manuscript (L169). We introduce the demonstration discriminator to mitigate this issue by downweighting the influence of the skill posterior outside the demonstration support.
> > We note that the skill posterior $q_\zeta$ should not be confused with the skill inference network $q_\omega$. The latter was trained with the task-agnostic offline data which has much wider support and thus works well even outside the demonstration support.
> >
> > &nbsp;
> >
> > **Discriminator potentially misspecified early in training**
> >
> > The discriminator $D(s)$ is trained fully offline _before_ RL training and then frozen during RL training; it is not trained adversarially (see page 5, footnote 2; we additionally explore full adversarial training in appendix, section E). Thus, it provides accurate guidance to the policy directly from the start of training.
> >
> > &nbsp;
> >
> > **Hard to assess completeness of the learned skills**
> >
> > Since we have access to demonstrations of the target task we can assess the quality of the learned skills before running RL, which is an advantage over prior skill-based RL works like Pertsch et al. 2020 [1] which did not have access to target task demonstrations: we can reconstruct the demonstration sequences using the pre-trained skills and use the resulting reconstruction error as a guide as to whether all relevant skills are represented in the learned skill space.
> >
> > &nbsp;
> >
> > We again thank you for your valuable feedback, please let us know if this addresses your concerns!
> >
> > &nbsp;
> >
> > **References**
> >
> > [1] Pertsch, Lee, Lim. "Accelerating reinforcement learning with learned skill priors." CoRL 2020.
> >
> > [2] Ross, Gordon, Bagnell. "A reduction of imitation learning and structured prediction to no-regret online learning." AISTATS 2011.
> >
> > [3] Mandlekar et al. "Roboturk: A crowdsourcing platform for robotic skill learning through imitation." CoRL 2018.
> >
> > [4] Gupta et al. "Relay policy learning: Solving long-horizon tasks via imitation and reinforcement learning." CoRL 2019.

---

> ### Author Response · Authors · 2021-08-28
> **Rebuttal Discussions**
>
> Dear reviewer fuNi,
>
> Please let us know if you have remaining concerns with regards to our submission following the rebuttal. The author discussion period ends this Monday, August 30.

---

### Official Review · Reviewer_5LPL · 2021-07-22

**Originality:** Good
**Technical Quality:** Very Good
**Clarity Of Presentation:** Very Good
**Impact:** 4

**Recommendation:**

Strong Accept: I recommend accepting the paper and will argue for my recommendation even if other reviewers hold a different opinion.

**Summary:**

This paper describes two mechanisms, which when combined can provide a good imitation learning algorithm to quickly learn a skill-based policy for a novel skill.
There are two main components:
A pretraining method (Skill representation Learning) for skill extraction from unstructured experience data.  This produces \pi(a|s,z) a skill-state-conditioned policy that will be used as the low-level controller later on.  This is learned through a variational learning objective with a regularization that forces the task embedding to be consistent during a particular trajectory.

A learning method train a policy purely from demonstrations (without a reward function) in a demonstration-data-efficient manner.  This method requires the training of:
D(s) - a demonstration/non-demonstration discriminator.
q_z(z|s), the task-based policy being learned.
P_a(z|s) - not entirely sure what this is, perhaps a model that provides z-probability conditioned on s_t but over the whole dataset? Or perhaps over a particular action sequence.... This could warrant being clarrified.

Once these are trained, a final policy pi_theta(z|s) is trained by maximizing a discriminator reward for a given task, and making sure that the policy is following the posterior predictor (q_z) when on-distrubition relative to task data, and following p_a(z|s) when out of distribution relative to task-specific data.

The authors show that this approach can find controllers that can complete more than 3 subtasks on average after 1-2 million training steps on the environment.  Thes tasks have no reward provided by the environment, and even if there was a sparse reward the exploration problem would render them unsolvable.
The authors show that given the various models above one can learn a new policy pi_theta(z|s) which can operate in the skill-space MDP to solve demonstration-defined tasks.

**Issues:**

1. Although overall the paper was easy to read, it would be very helpful to have a small table or some form of recap of all the different models: skill prior, skill posterier, skill inference model, policy etc. and what exactly each one learns.  I spent a lot of time going back and forth between sections to try to decipher what is going on.  Could you provide some form of recap before or around Eq. 4 to summarize all these, perhaps in some sort of table?

2. Small nit but on l. 162 you introduce the skill posterior but don't define the zeta subscript, is zeta a skill?  That was my deduction but this cost me some concentration points and could be easily clarified.

**Reviewer Expertise:**

Good: General knowledge of the area

**Strengths And Weaknesses:**

I found the paper pretty clear to read, straight-to-the-point, and the experiments quite convincing.  The method makes intuitive sense, and although it has a lot of moving parts each one seems to be well-justified and not overdone.  I also found the related work well put together and providing sufficient context to situate the proposed work relative to other papers.

My main comments on weaknesses would be that because there are a lot of moving parts, I had to go back and forth quite a bit, and I still have a couple elements that I'm not quite sure about (p_a(z|s) for example).  I've put specific fixes like this in the 'Issues' section.


**Summary Of Recommendation:**

I recommend acceptance, but please try and fix the issues mentioned in the next section.

---

> ### Author Response · Authors · 2021-08-23
> **Author Response (1/1)**
>
> Thank you for your detailed comments, we address your remarks below! We updated the manuscript and marked all changes made to address the reviewers’ comments in red.
>
> &nbsp;
>
> **Summarize model components in approach section**
>
> Thank you for this suggestion! As proposed, we added a paragraph before equation (4) in section 3.3 that lists all the components we pre-train from the offline data, which then get used in our target RL objective in equation (4). We agree with the reviewer that this makes parsing equation (4) much easier.
>
> &nbsp;
>
> **Undefined $\zeta$ subscript**
>
> Thanks for spotting this! The $\zeta$ subscript refers to the parameters of the skill posterior model. We clarified this in the text and also added descriptions for all other models for which the subscripts refer to model parameters.
>
> &nbsp;
>
> **Unclear what $p_a(z | s)$ refers to**
>
> We use $p_a(z | s)$ to refer to the learned skill prior, which is a state-conditioned distribution over latent skill embeddings $z$, trained from the whole task-agnostic dataset. It captures the distribution of skills that are meaningful to explore in a given state, irrespective of the task at hand (see also explanation in L149).
> Based on your remark we realized that the subscript “a” is unnecessary and now just refer to the skill prior as $p(z | s)$ in the updated manuscript for simplicity.
>
> &nbsp;
>
> Thanks again for your valuable feedback, please let us know if this addresses all your concerns!

---

### Official Review · Reviewer_RYNo · 2021-07-23

**Originality:** Good
**Technical Quality:** Good
**Clarity Of Presentation:** Good
**Impact:** 3

**Recommendation:**

Weak Accept: I recommend accepting the paper, but will not argue for my recommendation if the majority of other reviewers have a different opinion.

**Summary:**

The main contribution of the paper is to combine the skill-based RL with the demonstrations-based RL. The demo-based RL can suffer from the covariate shift problem. In contrast, the skill-based RL still needs to explore a relatively large options space. The idea is to combine both with a discriminator weight to avoid their disadvantages. Simply speaking, if the current state is covered by the demonstrations, the learned policy should behave like BC. If the current state is out of the demonstrations, the learned policy should behave like SPiRL.

**Issues:**

It would be good if the authors can compare the proposed method with SPiRL with the same set of skills. They should all depend on the states. If this is already the case, the authors should clarify it in the paper. (If I missed some sentences in the paper, please just correct me.)

I would suggest giving more details in the paper (but not only in supplementary materials) about the state and action definition for each experiment and what skills are learned. For example, for the maze navigation, it is not clear in the paper whether the goal position is in the state or not.

**Reviewer Expertise:**

Very good: Comprehensive knowledge of the area

**Strengths And Weaknesses:**

The idea is interesting and clearly explained in the paper. The experiments are thorough. However, in the paper, the skill is learned beforehand and fixed during the downstream training. I think that all three experiments can be considered standard RL problems with a small set of "actions".  These kinds of problems are difficult for the agents without considering the hierarchy of tasks.  But the experiments are only skill sequencing problems if skills are well trained. I noticed that the low-level policy described in the paper is dependent on the state. According to the authors, the one in SPiRL is, however, not dependent on the state. Is this also one reason why the proposed method outperforms SPiRL? Furthermore, compared to SPiRL, the proposed method still needs a set of demonstrations for different tasks.

**Summary Of Recommendation:**

Overall, the paper is well written. The idea is interesting. The question is whether this kind of method can be used or generalized for non-skill-sequencing problems. I think that it might be also difficult to extract skills automatically from demonstrations.

---

> ### Author Response · Authors · 2021-08-23
> **Author Response (1/1)**
>
> Thank you for your thoughtful comments, we hope we can address your concerns below!
>
> &nbsp;
>
> **Is SPiRL baseline conditioned on states too?**
>
> We indeed use the same learned skills for both, our algorithm and the SPiRL baseline, for fair comparison (see footnote 3 on page 7). As mentioned by the reviewer, these skills are conditioned on the environment state (the original SPiRL skill representation was unconditioned), which we found to improve performance for the SPiRL baseline as well (see also representation comparison in appendix, section C).
>
> &nbsp;
>
> **Add info about state / actions / skills to main paper?**
>
> Following the reviewer’s suggestion we added more detail about the state and action spaces in each of the tasks to the revised main paper (changes in section 4.1 marked in red) and also detailed the kinds of skills that are learned for each environment.
>
> &nbsp;
>
> **Is goal part of observation in maze?**
>
> The goal of the target maze navigation task is kept fixed, thus we do not need to include it in the agent’s observation. We clarified the components of the observations for the maze task in the revised paper L212.
>
> &nbsp;
>
> Thanks again for your valuable feedback, please let us know if this addresses all your concerns!

---

### Meta-Review · Area_Chair_tnEy · 2021-08-03

**Recommendation:** Accept (Poster)
**Confidence:** 4

**Metareview:**

=== comments before the discussion ===

The paper presents a demonstration-guided RL method, that leverages pre-trained skills.
In the initial review, the scores are split.
I recommend the authors to carefully address the concerns raised by reviewers. For example, following points seems to be important points:
- Reviewers 5LPL and fuNi shared the same impression that there are several moving parts in the proposed method.
- The reviewer fuNi raised concerns related to baselines in experiments, which is important for evaluating the method.
- The reviewer fuNi also raised concerns related to the out-of-distribution errors.

=== comments after the discussion ===

The author response clarified unclear points in the paper. Although there are several moving parts in the proposed method, the empirical results show the good performance in the kitchen task, and the pros outweigh the cons of the paper. Thus, the area chair recommends the acceptance of the paper.

---

> ### Author Response · Authors · 2021-08-23
> **Rebuttal Summary**
>
> Thank you for your summary of the most important points of the initial reviews! We addressed the concerns of each reviewer in detail in individual responses and updated the manuscript using their valuable feedback (changes marked in red). Below we briefly summarize our responses to each of the points you mentioned:
>
> - **many moving parts**: All components of our algorithm except for the target task policy are trained fully offline with simple supervised learning and thus converge stably and are robust to hyperparameter choices, contributing to our approach’s practicality. Following the reviewers’ suggestion we also included a summary of all pre-trained components in the approach section for better clarity.
> - **baseline performance**: We emphasize that our baselines match the performance reported in prior work wherever comparable, but in other cases are evaluated on substantially longer-horizon tasks than in prior works and thus struggle. We also point to qualitative analyses of baseline failures in L304 and Figure 8 of the original submission.
> - **out of distribution errors**: We clarify that the BC model the review refers to works well _within_ the support of its training data but, in line with results from the standard BC literature (Ross et al. 2011), suffers from distribution shift outside this support, which is expected. Our approach explicitly handles this distribution shift through the introduction of a learned support estimator, called “demonstration discriminator”.
>
> &nbsp;
>
> Please let us know if further clarifications are needed!

---

### Decision · Program_Chairs · 2021-09-13

**Decision:**

Accept (Poster)

**Comment:**

=== comments before the discussion ===

The paper presents a demonstration-guided RL method, that leverages pre-trained skills.
In the initial review, the scores are split.
I recommend the authors to carefully address the concerns raised by reviewers. For example, following points seems to be important points:
- Reviewers 5LPL and fuNi shared the same impression that there are several moving parts in the proposed method.
- The reviewer fuNi raised concerns related to baselines in experiments, which is important for evaluating the method.
- The reviewer fuNi also raised concerns related to the out-of-distribution errors.

=== comments after the discussion ===

The author response clarified unclear points in the paper. Although there are several moving parts in the proposed method, the empirical results show the good performance in the kitchen task, and the pros outweigh the cons of the paper. Thus, the area chair recommends the acceptance of the paper.